# Food and Nutrition in the Pathogenesis of Liver Damage

**DOI:** 10.3390/nu13041326

**Published:** 2021-04-16

**Authors:** Andrea Mega, Luca Marzi, Michael Kob, Andrea Piccin, Annarosa Floreani

**Affiliations:** 1Gastroenterology Department, Bolzano Regional Hospital, 39100 Bolzano, Italy; luca.marzi@sabes.it; 2Dietetics and Clinical Nutrition Unit, Bolzano Regional Hospital, 39100 Bolzano, Italy; michael.kob@sabes.it; 3Northern Ireland Blood Transfusion Service, Belfast BT9 7TS, UK; apiccin@gmail.com; 4Department of Internal Medicine V, Medical University of Innsbruck, A-6020 Innsbruck, Austria; 5Department of Industrial Engineering, University of Trento, 38100 Trento, Italy; 6Scientific Institute for Research, Hospitalization and Healthcare, 37024 Negrar-Verona, Italy; annarosa.floreani@unipd.it; 7Department Surgery, Oncology and Gastroenterology, University of Padova, 35128 Padova, Italy

**Keywords:** xenobiotics, micronutrients, P450 enzymes, vitamins, herbs, red vine, spices

## Abstract

The liver is an important organ and plays a key role in the regulation of metabolism and in the secretion, storage, and detoxification of endogenous and exogenous substances. The impact of food and nutrition on the pathophysiological mechanisms of liver injury represents a great controversy. Several environmental factors including food and micronutrients are involved in the pathogenesis of liver damage. Conversely, some xenobiotics and micronutrients have been recognized to have a protective effect in several liver diseases. This paper offers an overview of the current knowledge on the role of xenobiotics and micronutrients in liver damage.

## 1. Xenobiotics

The word “xenobiotics” is a large umbrella under which several chemical substances are identified. However, in physiological conditions, xenobiotics are not produced and therefore are not normally found within an organism. These foreign substances are usually toxic and may include carcinogenic agents, pharmaceutical compounds, pollutants, food preservatives, and others. Our organism can cathabolize xenobiotics through biochemical pathways which mostly operate within the hepatic system.

## 2. Hepatic Metabolism

The most important liver enzymes involved in the metabolism of foreign compounds (including drug and xenobiotics) can be divided into two groups: Phase I enzymes that catalyze the direct modification of the primary structure of compounds and Phase II enzymes that catalyze covalent binding (conjugation) to polar ligand, such as glucuronic acid, sulphate, glutathione, or amino acids. The conjugation phase enhances water solubility, allowing the xenobiotics to be excreted in the urine and, when excreted into the bile, limiting their reabsorption from the gut. The majority of *Phase I* metabolism is catalyzed by an important family of enzymes, the cytochromes P-450. These enzymes, within three distinct P450 gene families (CYP1, CYP2, CYP3), are important for the majority of *Phase I* metabolism of xenobiotics. Each family contains multiple members which are highly homologous to each other in terms of sequence of amino acids but differ in their ability to bind and metabolize specific xenobiotics. The P450 families are further divided into subfamilies, which share greater than 55% amino acid sequence homology. Subfamilies are defined with capital letters, such as CYP1A or CYP3A. Specific gene products are identified by Arabic numbers (i.e., CYP1A1 and CYP1A2), usually according to the order in which the specific P450 was discovered. Several substances contained in food can modulate the activity of CYPs (Table 1).

### 2.1. Vitamins

Interestingly, vitamins regulate CYPs in an important manner. In an elegant experimental study, Martini et al. showed that downregulation of P4502C11 in dietary-deficient mice was associated with a decreased level of serum androgen and retinol [1]. Conversely, dietary all-trans retinoic acid (ATRA) was able to sustain circulating androgen, but not retinol, concentrations. These data suggest that dietary vitamin A regulates P450 2C11 expression indirectly and that downregulation of the enzyme in dietary deficiency is a consequence of a decrease in circulating testosterone levels. In the liver, hepatocytes and hepatic stellate cells (HSCs) are involved in the metabolism of retinoids [2]. The hepatocyte plays an important role in the uptake and processing of dietary retinoid and in regulating the secretion of retinol-binding protein, which mobilizes hepatic retinoid stores. Altered metabolism of retinoids and consequent dysregulation of retinoic signaling in the liver contribute to hepatic disease [2]. In summary, activation of HSCs results in extracellular matrix deposition and the onset of liver fibrosis. Alcohol intake could induce abnormalities in the metabolism of retinoids in several ways: (i) competitive inhibition of the first step of retinoid oxidation catalyzed by alcohol dehydrogenase; (ii) accelerated metabolism of retinoic acid by inducing CYP enzymes, specifically, CYP2E1; (iii) enhanced retinol mobilization from the liver to peripheral tissues [3].

Vitamin A (vit A) deficiency impairs dark adaptation; conversely, vit A toxicity was described in patients taking large doses of vit A and in patients with type I hyperlipidemias and alcoholic liver disease [4]. In an anecdotal case study, a patient with intoxication due to an average intake of vit A of approximately 120 mg/day for at least 5 years developed an important chronic hepatic fibrosis, with liver biopsy showing fibrosis deposition around the central vein (personal communication). If we consider vitamin E, several studies have shown beneficial effects of vitamin E administration in patients with non-alcoholic steatohepatitis [5,6].

### 2.2. Components of Fruits and Vegetables

Natural products of certain fruits can interact with drugs affecting their metabolism and elimination. An example is grapefruit juice, which reduces enteric CYP3A4 activity, thus elevating the bioavailability of drug metabolism by enteric CYP3A4 [7]. The list of drugs includes highly prescribed medications such as digoxin, amitriptyline, warfarin, ethinyl estradiol. It is well known that the intake of fruit and vegetables is inversely correlated with insulin resistance-related diseases. In fact, fruits and vegetables provide protective dietary factors, such as antioxidant vitamins. In this view, the Mediterranean diet, which includes fruit and vegetables, is considered an excellent diet, useful for the prevention of NAFLD (Non-alcoholic fatty liver disease) [8]. A meta-analysis showed that an increase in the intake of vegetables, but not of fruit, determines a lower risk of hepatocellular carcinoma (HCC). The risk of HCC decreases by 8% for every 100 g/d increase in vegetable intake [9]. Another systematic search of the literature showed that an increase of the intake of vegetables was associated with a 39% reduction in HCC risk. A dose–response analysis showed that the risk of HCC was reduced by 4% with an increase of 100 g per day of intake. A subgroup analysis indicated that increased consumption of vegetables was associated with a 50% reduction of liver cancer risk in males but not in females. However, there was a non-significant association between fruit intake and HCC risk [10]. A 2017 meta-analysis summarized the findings on the relationship between the intake of grains, refined grains, vegetables, fruits, nuts, legumes, eggs, dairy products, fish, red meat, processed meat, and sugary drinks and the risk of mortality from all causes. An increased intake of whole grains, vegetables, fruit, dried fruit, and fish was associated with a decreased risk of all-cause mortality. Conversely, an increased intake of red meat and processed meat was associated with an increased risk of all-cause mortality [11]. Garlic is rich in plant elements with antioxidant and anti-inflammatory activity. A 2020 study of NAFLD patients showed that those taking garlic powder had lower levels of ALT (Alanine Aminotransferase), AST (Aspartate Aminotransferase), LDL–cholesterol (LDL–C), and triglycerides compared to those not taking it. [12]. Furthermore, 51% of the participants in the garlic group showed improvements in the severity of fat accumulation in the liver, compared to 16% in the control group. Another study involving over 24,000 people showed that men who consumed raw garlic more than seven times a week had an up to 29% reduced risk of developing fatty liver disease. However, this association was not observed in women. Additionally, raw garlic intake has been linked to a reduced risk of liver cancer [13].

### 2.3. Red Wine

The impact of red wine on the liver is very complex. Chronic ethanol intake is an important risk factor for the development of liver diseases. Ethanol and its bioactive metabolites can cause direct cytotoxic damage by acting as hepatotoxins [14]. The metabolism of ethanol can occur through both oxidative and non-oxidative pathways (Figure 1). Alcohol dehydrogenase (ADH) and acetaldehyde dehydrogenase (ALDH) are involved in the oxidative pathway [15]. The resulting end products are acetaldehyde, acetate, and high levels of nicotinamide adenine dinucleotide (NADH). Acetaldehyde causes liver damage by promoting inflammation and fibrogenesis, remodeling extracellular tissue [16], and causing apoptosis, leading to the production of immunogenic adducts in hepatocytes [17]. Cytochrome P450 2E1 (CYP2E1) is upregulated under conditions of ethyl abuse and helps ADH to convert alcohol into acetaldehyde [14]. Reactive oxygen species (ROS) generated by CYP2E1 are responsible for alcohol-related inflammatory damage. Bowel-derived lipopolysaccharide (LPS) also plays a role in fatty liver, inflammation, and fibrosis. In healthy persons, LPS from Gram-negative bacteria enters the portal circulation only minimally and is eliminated by resident macrophages and hepatocytes [18,19,20]. On the other hand, patients with a history of alcohol abuse present a damaged intestinal barrier, leading to a raised in permeability and to an increase in the levels of LPS. LPS will bind to the CD14 surface receptor on Kupffer liver cells via LPS-binding protein (LBP). This complex, via nicotinamide adenine dinucleotide phosphate (NADPH) oxidase, produces oxygen radical (ROS) and activates a receptor signaling cascade that favors the release of inflammatory cytokines, such as tumor necrosis factor alfa (TNF-α) [21,22]. TNF-α supports liver damage by increasing intestinal permeability and supporting necro-inflammatory liver damage [23] (Figure 1). Moderate red wine intake has been associated with hepatoprotective, anti-inflammatory, and lipid-regulating effects. These effects are mainly due to one of its polyphenolic compounds present in grape skin, named resveratrol [24]. Some studies hypothesize a protective effect of resveratrol in counteracting the oxidative effect of ethanol, thus affecting hepatic oxidative stress [25]. To date, demonstrating the role of moderate alcohol consumption in NAFLD patients remains a significant challenge. Furthermore, the association between alcohol abuse and the development of complications in chronic liver disease, including HCC, has led practitioners to recommend complete alcohol abstinence. Nevertheless, a 2018 study showed that modest wine consumption (1–70 g per week) in patients with NAFLD is associated with less fibrosis [24].

## 3. Herbal and Dietary Products

The use of plant and dietary supplements has a long tradition, especially in the Middle East and in South East Asia. Today, many people use botanical drugs to achieve better health, especially patients with abnormal liver blood tests. However, evidence of effectiveness is lacking. The effectiveness of the treatment is difficult to define, and there are doubts about the quality of the studies testing herbal remedies. Although herbal medicine has potential benefits in treating liver disease, its improper use can cause liver damage. Toxic liver damage from drugs or herbal supplements can be caused by inhibition of enzyme activity (e.g., cytochrome P450), induction of transcription of the P450 gene promoted by human pregnane X receptor (PXR), or interaction with prescribed drugs. Overall, the data on hepatotoxicity due to herbal supplements are limited and are mainly derived from case reports and series or retrospective databases. Data on hepatotoxicity are also limited due to the absence of regulations that do not require surveillance or reporting of adverse events as in the case of drugs (Table 2) [26,27].

Below we will analyze some studies that demonstrate a certain favorable effect on liver disease.

### 3.1. Milk Thistle

Silybum marianum has been used for more than two millennia as a food and as a medicinal herb for treatment of liver conditions. Silymarin is an extract of seeds, containing silybinin, silychristin, and silydianin. Milk thistle has antioxidant activity and reduces cholesterol, blood sugar, and blood pressure levels. Furthermore, milk thistle has anti-atherosclerotic and anti-obesity effects, as demonstrated by a review of several animal and human studies [28]. However, data are conflicting. For example, some studies have shown that silymarin can prevent the progression of alcoholic liver disease and increase survival, while other studies have not shown efficacy compared to placebo [29]. Although several studies exist, limitations are due to study designs, including different etiology and extent of liver disease, sample size, and variation in formulation, dosage, and therapy duration. The available studies are therefore not strong enough to support milk thistle as an effective product in liver disease. Silymarin is considered safe and not associated with negative side effects, even at high doses. Although silymarin presents low drug interactions and has no effect on cytochromes P-450, its use should be considered with caution, especially if sylmarin is taken in association with drugs with a narrow therapeutic window [30].

### 3.2. Ginseng

Ginseng is an herbal supplement known for its anti-inflammatory properties. Numerous studies conducted in vitro and in vivo have shown that ginseng has antioxidant properties and can reduce the damage caused by viruses, toxins, and alcohol. A 2020 study of 51 men who took ginseng documented a reduction in alanine aminotransferase and gamma-glutamyl transferase (GGT), compared to those who did not take it [31]. While these results are promising, further studies are needed to evaluate the direct action of ginseng on the liver. When used alone, ginseng is believed to be relatively safe for liver health, but when used in association with other medications, liver damage can occur.

### 3.3. Licorice

*Glycyrrhiza glabra* is a herb with powerful medicinal properties. Numerous scientific studies have shown that licorice root has anti-inflammatory and antiviral properties. Glycyrrhizin saponin is the main active ingredient of licorice root. A study of 66 people with fatty liver disease showed that taking 2 g of licorice root extract per day for 8 weeks significantly reduced cyto-necrosis markers, when compared to placebo [32]. In another small study, the assumption of vodka alone every night for 12 days compared to taking a licorice extract before vodka assumption, increased significantly liver damage [33].

### 3.4. Turmeric and Curcumin

Turmeric and curcumin can have several health benefits. Turmeric has powerful anti-inflammatory, antioxidant, and anticancer properties and therefore is a popular choice for liver disease prevention [34]. In a case–control study in people with NAFLD, it demonstrated a significant reduction in liver fat content and inflammatory marker levels [35]. Piperine is a compound contained in black pepper that improves the absorption of curcumin. Supplements with turmeric and curcumin are generally believed to be safe, although some cases of liver damage have been reported. It is unclear whether these cases occurred because of contamination of curcumin products or the products themselves [36].

### 3.5. Ginger

Ginger is used as a food and as a medicine. Ginger is known for its effects on nausea and vomiting. It is also used for other conditions, including liver disease [37].

Ginger can have also lipid-lowering effects, hypoglycemic effects, and antioxidant effects. A systematic review and meta-analysis showed that ginger may have a positive effect on lipid parameters, and a dose of ginger ≤2 g/day demonstrated a reduction in triglicerydes (TG) and total cholesterol (TC) levels [38].

A 3-month study in 46 people with NAFLD found that taking powdered ginger significantly reduced ALT, TC, LDL–C, blood glucose levels, HOMA (Homeostatic Model Assessment), C-reactive protein, compared to taking a placebo [39]. Similar results were presented in another study involving 44 NAFLD patients. Ginger intake showed a reduction in ALT, γ-GGT, inflammatory cytokines, HOMA, and fat accumulation in the liver and a significant effect on liver fibrosis and AST [40].

### 3.6. Dandelion

Dandelion (*Taraxacum officinale*) is an edible flowering plant of the Asteraceae family, known for its choleretic, diuretic, hypoglycemic, and anti-inflammatory properties [41]. Intake of dandelion leaf extract showed improved accumulation of liver lipids, TAG, TC, fasting blood glucose, and HOMA in mice with high-fat-diet-induced fatty liver disease [42]. Studies in mice also demonstrated a hepatoprotective effect of dandelion extracts against hepatotoxic substances (acetaminophen and ethanol), presumably by activating the NF-E2-related factor 2/heme oxigenase 1 (Nrf2/HO-1) pathways and inhibiting apoptotic pathways [43,44,45].

## 4. Spices

Several spices can inhibit drug metabolism enzymes and increase the plasma concentration of several drugs (Table 3) [46,47,48].

Many studies have documented the inhibition of CYP activity by spices or their constituents. However, a full evaluation of various spices has not been carried out to date. Kimura et al. studied the effects of 55 spices on CYP3A4 and CYP2C9 activity.

These enzymes are significantly inhibited by cinnamon, black or white pepper, ginger, mace, and nutmeg. Furthermore, the isolation of a new furan derivative produced by the fractionation of mace showed the most potent inhibitory activity of CYP2C9.

## 5. Other Alimentary Products

Some recent studies report the relationships between diet and NAFLD. Indeed, diet can play a key role in the development of non-alcoholic fatty liver disease. Below we analyze the association between some alimentary products and NAFLD (Table 4).

### 5.1. Legumes and Soy

Legumes (or pulses) are the seeds of various plants of the Leguminosae family. Common edible legumes include lentils, dry beans, soybeans, chickpeas, lupins, and peas. Legumes are very common around the world and are consumed either as whole seeds, or further processed (e.g., legume flour, tofu, tempeh, miso, bean sprouts, etc.). Legumes are significant sources of complex carbohydrates, proteins, dietary fibers, and minerals. Resistant starches in the seeds are converted into short-chain fatty acids (SCFA) by the gut microbiota, which serve as intestinal substrate for enterocytes and thus improve intestinal barrier integrity and prevent microbial translocation [49]. In animal models, legumes have been shown to have positive effects on glucose and lipid metabolism, both crucial features of the metabolic syndrome, mainly by upregulating genes related to beta-oxidation and acetyl-CoA-degradation and downregulating genes involved in glycolysis and lipogenesis [50]. In a small randomized controlled trial (RCT) of 42 premenopausal women with central obesity, a hypocaloric diet enriched in legumes showed a significant decrease in AST and ALT blood levels after 6 weeks, as well as better HOMA-IR (HOMA-Insuline Resistance), blood pressure values, triglyceride and fasting blood glucose levels compared to a hypocaloric diet without legumes [51]. A case–control study showed a significant association between lower risk of NAFLD in patients and greater intake of legumes (OR of NAFLD for total legumes 0.73; 95% CI 0.64–0.84, lentils 0.73; 95% CI 0.64–0.84% and beans 0.35; 95% CI 0.17–0.74) [52]. However, a metanalysis of three cross-sectional studies showed no significant association between legume consumption and the likelihood of NAFLD [53]. Soybeans (*Glycine max*) seems to play a special role in the prevention of NAFLD. Replacing a serving of soy with a serving of meat or fish was associated with 12–13% increased risk of fatty liver disease in a large Chinese cross-sectional study [54]. The protective effect is probably due in part to the high content of the protein β-conglycinin (7S globulin), which has been shown to downregulate the hepatic expression of PPARγ-2 in animal models [55]. In a randomized trial, daily administration of 5 g of β-conglycinin to subjects with hypertriglyceridemia resulted in a significant reduction in triglyceride levels and visceral adipose tissue after 12 weeks [56]. In a Japanese study, a high intake of miso, a fermented soy product, was associated with a decreased risk of liver cancer in men (HR 0.65; 95% CI, 0.48–0.89), while no association was observed for other soy products (fermented and unfermented) [57].

### 5.2. Fish and Vegetables Rich in Omega-3-Fatty Acids

Omega-3 fatty acids are polyunsaturated fatty acids, which are found either in sea fish and algae as long-chain eicosapentaenoic (EPA) and docosahexaenoic (DHA) acids or in certain nuts and seeds (walnuts, flax seeds, rapeseeds) as shorter-chain alpha-linolenic acid (ALA). Omega-3 fatty acids have numerous beneficial effects on lipid metabolism, insulin sensitivity, and glucose regulation, both in rodents and in humans [58]. A recent systematic review and meta-analysis showed that diets rich in omega-3 polyunsaturated fatty acids (PUFA) were associated with a reduced risk of metabolic syndrome (OR 0.74, 95% CI 0.62–0.89) [59]. The ratio of omega-3 to omega-6 fatty acids in the diet is of particular importance: the recommended omega-3/omega-6 ratio should be less than 1:4. In patients with NAFLD, an inverse correlation between omega-3/omega-6 ratio and histological degree of steatosis was found (r = 0.61, *p* < 0.001) [60]. In a small single-arm open trial, a low-omega-3/omega-6 ratio (1:4), normo-caloric diet ameliorated the metabolic phenotype of adolescents with fatty liver disease after 12 weeks [61]. Clinical trials examining the effects of omega-3 supplementation in NAFLD patients demonstrated reductions in triglycerides, liver enzymes, fasting blood glucose, and steatosis levels [62,63].

### 5.3. Extra Virgin Olive Oil

Extra virgin olive oil (EVOO) has several effects on the liver, reducing fatty liver, swelling of hepatocytes, fibrogenesis and preventing lipid peroxidation, thanks to its high levels of monounsaturated fatty acids. A published study concluded that the most relevant effects of EVOO are activation of nuclear transcription factors and preventing cellular inflammatory response, endoplasmic reticulum stress, autophagy, and lipogenic response [64].

### 5.4. Beverages Containing Caffeine

Caffeine is a xanthine alkaloid and the most important component of coffee, tea, and chocolate. Some studies suggest that regular caffeine intake may have protective effects on the progression of chronic liver disease and the development of liver cancer. Furthermore, high doses have not been associated with liver damage but with impaired brain, heart, and muscle function. Caffeine has many anti-inflammatory and immunomodulatory effects, but high-caffeine energy drinks can cause liver damage such as acute liver necrosis or ischemic hepatitis [65,66].

#### 5.4.1. Coffee

Coffee is one of the most consumed beverages in the world and has several properties: antimicrobial, prebiotic, anti-inflammatory, antioxidant, anti-lipidemic, anti-obesity, anti-diabetes activity and cardiovascular protective properties. Several studies have also shown how coffee intake can reduce the incidence of liver disease, including fibrosis, cirrhosis, and cancer and even overcome all-cause mortality and suicide risks [67].

A Scottish study found that coffee consumption was associated with a reduced prevalence of cirrhosis in patients with chronic liver disease, regardless of the amount of coffee [68].

Experimental and clinical evidence suggests that coffee consumption has also protective effects against metabolic syndrome (MS). Hino et al. showed that coffee intake is associated with a lower incidence of MS [69]. Similar results were published by Catalano et al., which showed that coffee drinkers have a lower severity of fatty liver, including obesity and insulin resistance. Subsequently, a series of clinical trials around the world have confirmed these studies, showing that coffee protects against MS as well as NAFLD/NASH [70,71,72,73,74]. In addition, there is growing evidence of the association between coffee consumption and the risk of HCC. Four meta-analyses conclude that prospective cohort and case–control studies showed an inverse relationship between coffee intake and HCC [75,76,77,78]. In summary, summary relative risk (RRs) for HCC were, respectively, 0.66, 0.78, and 0.50 for regular, low, and high coffee consumption, while it was 0.85 for each cup increase per day [77]. Another meta-analysis confirmed these data, showing that each cup-a-day increase of coffee was associated with a 15% reduction in liver cancer risk, while decaffeinated coffee consumption did not demonstrate a significant decrease in liver cancer [78].

These results could be determined by the antiproliferative, antioxidant, antifibrotic, or proapoptotic effects of coffee blends or major bioavailable coffee compounds, as documented by in vitro studies.

#### 5.4.2. Green and Black Tea

Tea is a commonly consumed beverage in Asia and is believed to have healthful properties, and its ingredients have chemoprotective, antiproliferative, and antioxidant action. Human clinical studies showed that 1.6 g of green tea extract are well tolerated, while 9.9 g per day are the maximum tolerated dose in humans. Although evidence of the effect of green tea consumption on HCC risk remains ambiguous, a recent meta-analysis concluded that increasing green tea intake may have a preventative effect against liver cancer [79]. A review summarized several animal and human studies of green tea on NAFLD, suggesting that tea prevents this disease [80]. Although tea may have chemoprotective, antiproliferative, and antioxidant properties and is present in several products, it is not approved for any medical indication. The hepatotoxicity highlighted by preclinical and human data seems to be caused by catechin, a component of green tea, as well as by immunological mechanisms, as demonstrated by the association with the HLA B * allele 35: 01.

#### 5.4.3. Chocolate

A study published in 2016 and conducted on a population in Luxembourg suggested that the daily consumption of chocolate can improve liver enzymes and protect against insulin resistance [81]. Another cross-sectional study suggested that cocoa polyphenols may exert antioxidant activity in patients with NASH [82]. The same group showed that cocoa polyphenols can also improve the endothelial function in patients with NASH [83].

#### 5.4.4. Lemon Juice

Lemon juice (LJ) is a widely consumed drink. A study in mice showed that LJ inhibits the alcohol-induced increase in ALT, AST, hepatic TG, and lipid peroxidation levels. Lemon juice also has a protective effect on alcohol-induced liver damage, according to the histological analysis of liver tissue in mice [84].

## 6. Discussion and Conclusions

The liver is susceptible to a multitude of injuries that can cause liver damage, leading to chronic liver disease, cirrhosis, and HCC. Many foods/products are involved or could be involved in this process [85]. The use of plants and the consumption of fruit have played a fundamental role in human health care, and several scientific studies have identified chemicals called phytochemicals with beneficial effects. Research conducted on certain foods, fruits, and plants, frequently consumed by humans, has shown hepatoprotective actions.

Despite enormous advances in experimental and clinical research, there are no fully effective drugs that stimulate liver regeneration by offering restoration of liver function when impaired. It is therefore necessary to identify more effective and less toxic pharmaceutical alternatives for the treatment of liver disease. The associations between food and drink and the development of chronic diseases are the subject of nutritional research. In the literature, pooled/meta-analyses and systematic reviews (PMASRs) aim to better define these associations.

This brief review confirms that plant foods are more protective against chronic liver disease than animal foods. Among plant foods, those based on cereals are more protective than fruits and vegetables. Animal foods such as dairy or dairy products have no effect on the risk of developing chronic liver disease, while red or processed meats tend to increase the risk.

Among beverages, tea was the most protective, and soft drinks the least protective against liver diseases. The conclusion is that there is a need to further study the associations between food and drink groups and diseases of the digestive system, in particular, liver diseases [86].

In clinical practice, hepatological recommendations are often based on promoting some foods and discouraging the use of others. The so-called “Mediterranean diet” is strongly recommended. The Mediterranean diet (MD) includes grains, vegetables, and fruits, olive oil, nuts, fish, white meat, and legumes in moderation. In fact, MD has been shown to decrease cardiovascular disease, metabolic syndrome, and type 2 diabetes. Although MD appears attractive for its potential to improve liver status, the literature on the efficacy of such a diet is still limited.

Though plant and dietary supplements are used all over the world, there is a tendency to underestimate their intake by patients, as well as by physicians, especially non-hepatologists. In fact, it is not uncommon to diagnose herbal hepatotoxicity. The incidence and precise manifestations have not been well characterized. Even on the use of caffeinated beverages, herbal and dietetic products for the prevention or treatment of liver disease, a full consensus from scientific societies is lacking. The problem with plant consumption lies in the limited availability of prospective observational and randomized clinical trials on plant safety for long-term and large-scale use. Indeed, well-designed randomized controlled trials are needed to confirm and to understand the role of these substances for the prevention and/or for treatment of liver disease. Limitations are due to the study designs used, such as etiology and extent of liver disease, sample size, and variations in formulation, dosage, and duration of therapy. Many herbs and plants have been indicated as a major cause of liver injuries. However, the toxic compounds remain to be determined, and most of the causal relationships between these products and hepatotoxicity are unconfirmed and lack convincing evidence [86,87].

## 7. Future Directions

Overall, our current knowledge on xenobiotics and liver toxicities is limited. Similarly, the true potential of some nutrients (e.g., curcuma) in protecting endothelial cells and preventing cellular damages is poorly understood and often unknown. For these reasons, we believe that further accurate studies and classification of xenobiotics are required and should be strongly encouraged. In the future, it will be useful to analyze the inter-individual differences using big data analytics and artificial intelligence to provide tailored, individualized nutritional guidelines.

## Figures and Tables

**Figure 1 nutrients-13-01326-f001:**
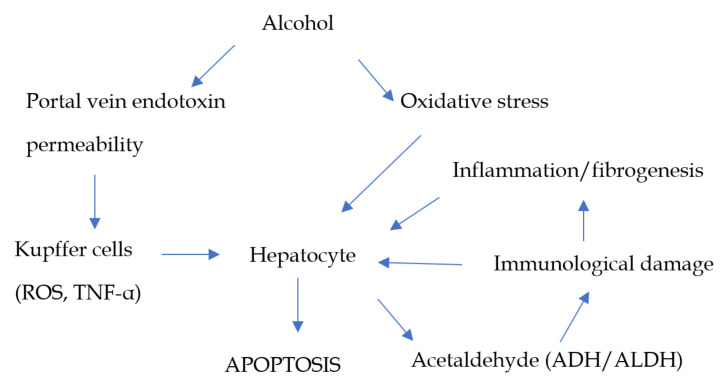
Main steps of alcohol metabolism in the liver and liver damage. ROS, reactive oxygen species, TNF-α, tumor necrosis factor α, ADH, alcohol dehydrogenase, ALDH, acetaldehyde dehydrogenase.

**Table 1 nutrients-13-01326-t001:** Substances modulating cytochromes P450 (CYPs) activity.

Vitamins
Components of fruit/vegetables
Red wine
Herbs
Spices

**Table 2 nutrients-13-01326-t002:** Interactions between herbal supplements and P450 enzymes.

Herbal Supplements	Compound	Effect on Enzyme
Milk Thistle (*Silybum marianum*)	Silybin	Inhibition of CYP3A4
Chinese ginseng	Extract	Activation of CYP3A4 promoter via hPXR
Licorice	Extract	Activation of CYP3A4 promoter via hPXR
Ginger	Extract	Inhibition of CYP2C9 Inhibition of CYP3A4
Tumeric (*Curcuma longa*)	Not noted	Inhibition of CYP2C9 Inhibition of CYP2C19 Inhibition of CYP2D6 Inhibition of CYP3A4
Tumeric (*C. longa*)	Curcumin	Activation of CYP3A4 promoter via hPXR

**Table 3 nutrients-13-01326-t003:** Interactions between spices and P450 enzymes.

Spice	Compound	Targeted Enzyme
Chilli pepper	Capsaicin	CYP1A CYP2B CYP2E1
Black pepper	Piperine	CYP1A CYP3A4
Cloves, nutmeg, cinnamon	Methylenedioxyphenyl	CYP1A2 CYP2E1 CYP3A4

**Table 4 nutrients-13-01326-t004:** Alimentary products and NAFLD.

Legumes and soy
Fish and vegetables rich in omega-3-fatty acids
Extra virgin olive oil
Beverages containing caffeine (coffee, green and black tea, chocolate)
Lemon juice

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
