# Peer review of "Food and Nutrition in the Pathogenesis of Liver Damage"

_nutrients, 2021, doi:10.3390/nu13041326_

Round 1

Reviewer 1 Report

Reviewer’s Comments:

The manuscript " Food and Nutrition in the Pathogenesis of Liver Damage " by Mega et al reviews the role of xenobiotics and micronutrients in liver damage.

Major comments:

  1. The definition of xenobiotics in the first line of the ‘Introduction’ section i.e., “Xenobiotics are components of the diet that are subjected to liver metabolism” is incorrect. Xenobiotics are chemical substances that are foreign to the body. Please rewrite the definition of xenobiotics and add a separate paragraph that briefly describes xenobiotics in terms of their accumulation and metabolism.
  2. Please cite table #1 in the text.
  3. Please provide a reference(s) for the last line of the conclusions section – “Herbal hepatotoxicity has been reported many times affecting many herbs and plant products, but most of the causal relationships are unconfirmed and lack convincing evidence”.
  4. Please add a brief paragraph on “future directions” at the end of the discussion/conclusions section.
  5. Please be consistent with the style of references. For example, in references #4 and #7, pages numbers are presented in different formats.
  6. Please proofread for spelling and grammatical errors.

Author Response

  1. The definition of xenobiotics in the first line of the ‘Introduction’ section i.e., “Xenobiotics are components of the diet that are subjected to liver metabolism” is incorrect. Xenobiotics are chemical substances that are foreign to the body. Please rewrite the definition of xenobiotics and add a separate paragraph that briefly describes xenobiotics in terms of their accumulation and metabolism. Answer: The definition of “xenobiotics” has been corrected in the paper, we’ve added a separate paragraph on xenobiotics. Thank you very much for your observation.
  2. Please cite table #1 in the text. Answer: Table 1 has been cited in the text, according to your suggestion
  3. Please provide a reference(s) for the last line of the conclusions section – “Herbal hepatotoxicity has been reported many times affecting many herbs and plant products, but most of the causal relationships are unconfirmed and lack convincing evidence”. Answer: we’ve added 2 references for the cited text (86 and 87). 86. Ki Tae S., Dong Joon K. Drug-induced liver injury: present and future. Clin Mol Hepatol. 2012 Sep; 18(3): 249-257. Published online 2012 Sep 25. doi:3350/cmh.2012.18.3.249. 87. Amadi Nwadiuto C., Orisakwe Orish E. Herb-induced Liver Injuries in Developing Nations: An Uptodate. Toxics 2019 Jun; 6(2): 24. Published online 2018 apr 17. doi 3390/toxics6020024
  4. Please add a brief paragraph on “future directions” at the end of the discussion/conclusions section. Answer: we provided a brief paragraph on “future directions” at the end of the paper, after the conclusions 
  5. Please be consistent with the style of references. For example, in references #4 and #7, pages numbers are presented in different formats. Answer: all the references have been reviewed and standardized according pubmed layout, according to your suggestion
  6. Please proofread for spelling and grammatical errors. Aswer: the text has been reviewed, and text and grammatical errors corrected

Reviewer 2 Report

This review on the relationship between foods and liver diseases, in particular non-alcoholic fatty liver disease and hepatocellular carcinoma, is clear and up-to-date.

The caution of the Authors in judging the controversial data of the literature is appreciable.

Studies more rigorous and based on large  sample size are strongly needed.

Author Response

Thank you very much for your comments.  In the future, it will be useful to analyze the inter-individual differences using big data analytics and artificial intelligence to find out causal relationships between xenobiotics and hepatotoxicity and to  provide tailored, individualized nutritional guidelines. Thank you. 

Round 2

Reviewer 1 Report

Please replace "These foreign substances are" with  "Xenobiotics are foreign susbstances that are" in the first paragraph i.e., "1. Xenobiotics".